

# Effects of functional correction training on injury risk of athletes: a systematic review and meta-analysis

Junxia Chen, Chunhe Zhang, Sheng Chen and Yuhua Zhao

P.E School, Hubei University, Wuhan City, Hubei Province, P.R. China

## ABSTRACT

**Background**. We explored functional correction training using the Functional Movement Screen (FMS^TM) tool. We also analyzed the effects of training on the injuries of athletes in a systematic review and meta-analysis of non-randomized clinical trials.

**Methodology**. We collected twenty-four articles from PubMed, CENTRAL, Scopus, ProQuest, Web of Science, EBSCOhost, SPORTDiscus, Embase, WanFang, and CNKI that were published between January 1997 to September 2020. Articles were selected based on the following inclusion criteria: randomized and non-randomized controlled trials, studies with functional correction training screened by FMS^TM as the independent variable, and studies with injury risk to the athlete as the dependent variable. Data conditions included the sample size, mean, standard deviation, total FMS^TM scores, number of injuries, and asymmetry movement patterns after interventions in the experimental and control groups. Exclusion criteria included: conference abstracts, cross-sectional studies, articles with retrospective study design.

**Results**. Twelve non-randomized trials were included in the meta-analysis. The injury risk ratio of athletes after functional correction training was 0.39 RR (95 CI [1.50–1.93]; $Z = 15.53; P < 0.0001; I^2 = 2.6\%$), indicating an improvement of athletes functional patterns.

**Conclusion**. Grade B evidence indicates that functional correction training based on FMS^TM may improve the functional patterns of athletes and Grade D evidence indicates that it may reduce the risk of sports injury. However, the true effect is likely to be different from the estimate of the effect. Therefore, further studies are needed to explore the influence of functional correction training on the injury risks of athletes. Protocol registration: CRD42019145287.

## INTRODUCTION

The mechanisms of sports injuries in athletes are complex and multifactorial with many potential risk factors for increasing the risk of injury. FMS^TM is used to evaluate the basic sports patterns of athletes and to screen potential risk factors for injury. It comprises seven basic movements: active straight leg raise, shoulder mobility, trunk stability push-up, trunk rotary stability, in-line lunge, hurdle step and deep squat. Each movement is scored on a scale of 1–3 for a total score of 21 points (*Cook, Burton & Hoogenboom, 2006a*; *Cook, Burton & Hoogenboom, 2006b*). FMS^TM assists in program design by systematically

Corresponding author
Junxia Chen, 775100275@qq.com

using corrective exercises to normalize or improve fundamental movement patterns (*Cook, 2011*; *Cook et al., 2014a*; *Cook et al., 2014b*). An earlier study reported that injury prevention and performance enhancement programs should consider including FMS[TM] or a similar movement screening tool and their associated exercises to normalize dysfunctional movement with the goal of injury reduction and performance improvement (*Kiesel, Plisky & Butler, 2011*). Therefore, it is important to evaluate the relationship between functional correction training after FMS[TM] of athletes and sports injuries.

Functional correction training after FMS[TM] has shown inconsistent results when reported in other populations, including a randomized controlled trial of patients undergoing anterior cruciate ligament reconstruction in whom a set of gradually progressive functional corrective training exercises may significantly improve the function and movement of the knee joint (*Chao et al., 2018*). Several non-randomized controlled trials of firefighters have shown that personalized corrective exercises can improve FMS[TM] scores (*Basar, 2017*; *Jafari, Zolaktaf & Ghasemi, 2019*). A series of studies by *Frost et al. (2012)*, *Frost et al. (2015a)*, *Frost et al. (2015b)* and *Frost et al. (2015c)* reported that the effectiveness of FMS[TM] training requires the consideration of various factors, such as the number and type of participants, the scoring method (paper or video), the feedback provided during the test, and supervision by a coach. FMS[TM] may not be a viable tool to assess movement behaviors regardless of whether it is graded qualitatively using composite or task scores or quantitatively via kinematic analyses (*Frost et al., 2017*; *Cornell, 2016*) and the effect of functional correction training on firefighters after FMS[TM] was unclear.

Several non-randomized controlled studies of the functional correction training of athletes (*Kiesel, Plisky & Butler, 2011*; *Kiesel, Butler & Plisky, 2014*; *Bayati et al., 2019*; *Campa, Spiga & Toselli, 2019*; *Riela & Bertollo, 2019*; *Kovac, 2018*) have reported that it may improve their FMS[TM] scores as well as reduce asymmetry in functional patterns. Additional studies (*Xuhua & Ye, 2015*; *Dinc et al., 2017*; *Hui & Baoai, 2019*) have reported that athletes had significantly improved FMS[TM] scores and reduced sports injuries. The training of athletes' functional correction after FMS[TM] may have been effective. However, these findings must be verified since the studies included small sample sizes, some had no control group, and they lacked a strict randomized control design.

The summarized results of systematic reviews and meta-analyses of the total FMS[TM] score of a mixed population to predict the sports injury risk are contradictory and they do not support the predictive validity of FMS[TM] (*Dorrel et al., 2015*; *Moran et al., 2017*). However, other studies have reported that participants with composite scores equal to or less than 14 had a significantly higher likelihood of an injury compared to those with higher scores (*Bonazza et al., 2017*). Individuals classified as high risk by FMS[TM] are 51% more likely to be injured than those classified as having a low risk (*Santos Bunn, Rodrigues & Da Silva, 2019*). Two studies pertaining to FMS[TM] and sports injuries of athletes considered the total scores and asymmetry of the FMS[TM] to be more useful for evaluating the injury risk of older athletes (*Moore et al., 2019*). Another review reported that the relationship between the FMS[TM] score and injury is unclear as the heterogeneity of the study populations (type of athletes, age, and sport exposure) and the definition of injury used in the studies made it difficult to draw definitive conclusions (*Trinidad-Fernandez, Gonzalez-Sanchez*

& Cuesta-Vargas, 2019). Therefore, there is no clear conclusion about the relationship between FMS[TM] and sports injuries.

Some reviews that analyzed the effects of functional correction training reported that the plan was effective and improved the limitations of exercise patterns (*Minthorn et al., 2015*; *Kraus et al., 2014*). There is currently no meta-analysis or systematic review of this topic, and establishing the impact of functional correction training on sports injuries of athletes is a challenging and important task. We sought to explore the impact of functional correction training after FMS[TM] screening on the injury risk of athletes and to determine whether functional correction training after FMS[TM] screening could increase total FMS[TM] scores and reduce the incidence of asymmetry in movement patterns of athletes. We hypothesize that functional correction training after FMS[TM] may reduce the sports injury risk in athletes.

## MATERIALS AND METHODS

### Agreement and registration
The systematic review and meta-analysis were performed in accordance with the preferred reporting items for systematic reviews and meta-analyses. We followed the Cochrane Collaboration Handbook while conducting our research (*DerSimonian & Kacker, 2007*; *Moher et al., 2009*). This systematic review does not include individual patient data; therefore, ethical approval was not required. The research was registered in PROSPERO (Registration no. CRD42019145287).

### Inclusion and exclusion criteria
Participants: Adolescent and adult athletes; males and females.
Interventions: After initial FMS scores were obtained, athletes were prescribed an individualized FMS-score-based training program designed to correct the identified movement deficits. The training program included self-administered trigger point treatments, self and partner stretching of major muscle groups, and strength and stability exercises. The intervention plans were conventional training and functional correction training.
Comparator: Studies were required to have a comparator group that performed conventional training only.
Outcomes: Data indicators were sample size, mean, standard deviation, total FMS[TM] scores, number of athletes with sports injuries, and functional pattern asymmetry after intervention of the experimental and control groups.
Types of studies to be included: Randomized and non-randomized controlled trials were included.
Exclusion criteria: Conference abstracts, cross-sectional studies, and retrospective studies were excluded.

### Article sources, retrieval, and selection
Two authors (S Chen) and (Y Zhao) independently searched the literature. Disagreements over the validity of the findings were solved through consensus and by discussion with

a reference author (C Zhang). Ten electronic databases including PubMed, CENTRAL, Scopus, ProQuest, Web of Science, EBSCOhost, SPORTDiscus, Embase, Wanfang, and CNKI were searched for full texts published between January 1997 and September 2020. The following search terms and MeSH terms were used: functional movement screen OR fms* OR functional movement screen* AND injury* OR injury prediction OR injury risk OR injury prevention screening OR athletic injuries [MeSH] AND functional training OR functional correction training OR corrective exercise training AND sport* OR athlete*OR player. Articles written in Chinese were limited to full text. The Chinese version of key words "FMS$^{TM}$, functional training and athletes" were also used. Additionally, the references of the selected articles were searched manually to obtain other potentially related studies. Table 1 shows the systematic search strategy.

## Data extraction and collection procedure

All duplicates were removed before our two investigators (S Chen, Y Zhao) independently screened the titles and abstracts for eligibility. Two investigators independently assessed the full text of the remaining articles for eligibility. The resulting differences were resolved by a reference author (C Zhang), Results provided by each investigator were compared after each stage, and any discrepancies were resolved by discussion. The following data were extracted from the original reports: authors, year, and publication; country; sample characteristics (sample size, age, and sex); functional correction training program; conventional training program; and main results (average values and standard deviations), including total FMS$^{TM}$ scores, number of athletes with sports injuries, and functional movement asymmetry after intervention in the experimental group and the control group.

We defined musculoskeletal injuries as sports injuries and they were considered the main outcome as to whether our intervention test reduced the risk of sports injuries. Other additional outcomes were total FMS$^{TM}$ score and functional movement asymmetry. Of these, bilateral muscle function asymmetries through FMS$^{TM}$ were defined as functional movement asymmetry of the six basic movements: active straight leg raise, shoulder mobility, trunk rotary stability, in-line lunge, hurdle step and deep squat.

## Quality evaluation

The risk of bias in non-randomized studies was assessed in a manner similar to that used for randomized trials as recommended by the Cochrane Collaboration Handbook for bias assessment of non-randomized studies (*Higgins & Green, 2011*). Two researchers were asked to independently evaluate the quality of articles according to 11 factors of the PEDro quality score scales (*PEDro, 2021*, https://www.pedro.org.au). The Spearman rank correlation coefficient was calculated to determine inter-rater reliability of the two researchers (Spearman's rho = 0.779), and a strong level of agreement was found.

The systematic error of 15 articles was assessed using Cochrane's risk of bias tool (RevMan; *Cochrane, 2020*). The same researchers independently scored each trial for the risk of bias. In the case of disagreement, a third researcher assessed the questionable item, and agreement was sought by consensus. Each study was graded for the following domains: random sequence generation, allocation concealment, blinding of participants

**Table 1  Search history.**

**PubMed up to September 2020**

Search: (((((functional movement screen) OR (fms*)) OR (functional movement screen*)) AND (((((injury*) OR (injury prediction)) OR (injury risk)) OR (injury prevention screening)) OR (Athletic injuries[MeSH]))) AND ((functional training) OR (corrective exercise training) OR(functional correction training))) AND ((sport*) OR (athlet*)OR(player)) Filters: Free full text, Full text, from 1997 –2020

**Scopus up to September 2020**

Search: (((((functional movement screen) OR (fms*)) OR (functional movement screen*)) AND (((((injury*) OR (injury prediction)) OR (injury risk)) OR (injury prevention screening)) OR (Athletic injuries[MeSH]))) AND ((functional training) OR (corrective exercise training) OR(functional correction training))) AND ((sport*) OR (athlet*)OR(player)) Filters: Free full text, Full text, from 1997 –2020

**Scopus up to September 2020**

TITLE-ABS-KEY("functional movement screen")OR("fms*")OR("functional movement screen*")AND("injury*")OR("injury prediction")OR("injury risk")OR("injury prevention screening")OR("Athletic injuries exp")AND("functional training")OR("corrective exercise training")OR("functional correction training") AND("sport*")OR("athlet*")OR("player")

**EMbase up to September 2020**

1 "functional movement screen" or "fms* af' or "functional movement screen* ".af.
2 "injury*" or "injury prediction" or "injury risk" or "injury prevention screening" or "Athletic inj uries exp".af.
3 "functional training" or "corrective exercise training" or "functional correction training".af.
4 "sport*" or "athlet*" or "player".af.
5 "functional movement screen" or "fms* af' or "functional movement screen* af "and "injury*" or "injury prediction" or "injury risk" or "injury prevention screening" or "Athletic injuries exp" and "functional training" or "corrective exercise training" or "functional correction training" and "sport*" or "athlet*" or "player". af.

**Web-sicence up to September 2020**

# 5 #4 AND #3 AND #2 AND #1
# 4 TS=(sport*) OR TS= (athlet*) OR TS=(player)
# 3 TS=(functional training) OR TS= (corrective exercise training) OR TS=(functional correction training)
# 2 TS=(injury*) OR TS= (injury prediction) OR TS=(injury risk) OR TS= (injury prevention screening) OR TS= (Athletic injuries[MeSH])
# 1 TS=(functional movement screen) OR TS= (fms*) OR TS= (functional movement screen*)

**EBSOhost up to September 2020**

S1 ((functional movement screen) OR (fms*)) OR (functional movement screen*)
S2 ((((injury*) OR (injury prediction)) OR (injury risk)) OR (injury prevention screening)) OR (Athletic injuries[MeSH])
S3 (functional training) OR (corrective exercise training) OR (functional corrective training)
S4 (sport*) OR (athlet*) OR (player)
S5(((((functional movement screen) OR (fms*)) OR (functional movement screen*)) AND (((((injury*) OR (injury prediction)) OR (injury risk)) OR (injury prevention screening)) OR (Athletic injuries[MeSH]))) AND (((functional training) OR (corrective exercise training) OR (functional correction training)))) AND (((sport*) OR (athlet*) OR (player))

**CENTRAL up to September 2020**
**ProQuest**- Dissertations & Theses
**Proquest-Health & Medical Collection**

1 "functional movement screen" OR "fms*"OR "functional movement screen"
2 "injury*" OR "injury prediction" OR "injury risk" OR "injury prevention screening" OR "Athletic injuries[MeSH]"
3 "functional training" OR "corrective exercise training" OR "functional corrective training"
4 "sport*" OR "athlet*" OR "player"
5 "functional movement screen"OR"fms* " OR"functional movement screen* "AND"injury*" OR"injury prediction" OR "injury risk" OR "injury prevention screening" OR "Athletic injuries[MeSH] " AND "functional training" OR "corrective exercise training" OR "functional correction training" AND "sport*" OR"athlet*" OR "player"

SPORTDiscus **up to September 2020**

S1 ((functional movement screen) OR (fms*)) OR (functional movement screen*)
S2 ((((injury*) OR (injury prediction)) OR (injury risk)) OR (injury prevention screening)) OR (Athletic injuries[MeSH])
S3 (functional training) OR (corrective exercise training) OR (functional corrective training)
S4 (sport*) OR (athlet*) OR (player)
S5(((((functional movement screen) OR (fms*)) OR (functional movement screen*)) AND (((((injury*) OR (injury prediction)) OR (injury risk)) OR (injury prevention screening)) OR (Athletic injuries[MeSH]))) AND (((functional training) OR (corrective exercise training) OR (functional correction training)))) AND (((sport*) OR (athlet*) OR (player))

CNKI **up to September 2020**

FMS, functional training, athletes

WANFANG **up to September 2020**
FMS, functional training, athletes

and personnel, blinding of outcome assessment, incomplete outcome data, selective reporting, and other bias. Each domain was rated as having a low or high risk of bias. In the case of insufficient reported information or information with a questionable interpretation that was unclear, the risk of bias for this item was rated as unclear.

## Effect index and data aggregation method

Meta-analyses were performed with R3.3.2. The random effects method or the fixed-effects method was used depending on the heterogeneity (*DerSimonian & Kacker, 2007*). The risk ratio (RR) was used to combine the athletes' sports injuries and asymmetry of functional patterns after intervention. The mean difference (MD) was used to combine the athletes' total FMS$^{TM}$ scores. A 95% confidence interval (CI) was also used. The effect sizes of the results were evaluated as follows: large effect size, >0.8; medium effect size, 0.5–0.79; and small effect size, 0.00–0.49 (*Higgins & Green, 2011*). The heterogeneity of results across studies was evaluated using the $I^2$ statistic as follows: may not be important, 0–40%; moderate heterogeneity, 30–60%; substantial heterogeneity, 50–90%; and considerable heterogeneity, 75–100% (*Higgins & Thompson, 2002*). Additionally, the adopted significance level was P ≤0.05. The publication bias was tested using Egger's linear regression (*Sterne, Egger & Smith, 2001*). Finally, a sensitivity analysis was performed by eliminating the research literature item-by-item and calculating the combined value of the remaining literature to determine if the results changed.

## Level of evidence

The quality of the evidence associated with the meta-analysis results was assessed using the Grading of Recommendations Assessment, Development and Evaluation approach (GRADE) (*Guyatt et al., 2011a*; *Guyatt et al., 2011b*; *Guyatt et al., 2011c*; *Guyatt et al., 2011d*; *GRADEproGDT, 2020*).

# RESULTS

## Study selection

We excluded 594 of the 696 articles identified in the initial literature search. Further screening was conducted according to the aforementioned inclusion criteria and quality assessments. Discrepancies were resolved through third-party mediation. Twenty-four articles met the inclusion criteria and included in the systematic review and 12 were selected for this meta-analysis. None of the included studies were a randomized controlled trial. Figure 1 shows the systematic search strategy and selection process.

## Study characteristics

The study included a total of 538 participants; 258 were included in the experimental group and 280 in the control group. Detailed information regarding the training status is shown in Tables 2–4. The age of the participants ranged from 9.6 to 26.5 years; the average ages of the experimental group and control group were 18.56 ± 4.17 years and 19.04 ± 4.92 years, respectively. The shortest experiment time was 6 weeks and the longest was 20 weeks. The average experiment time was 9.33 ± 4.32 weeks. The shortest intervention frequency

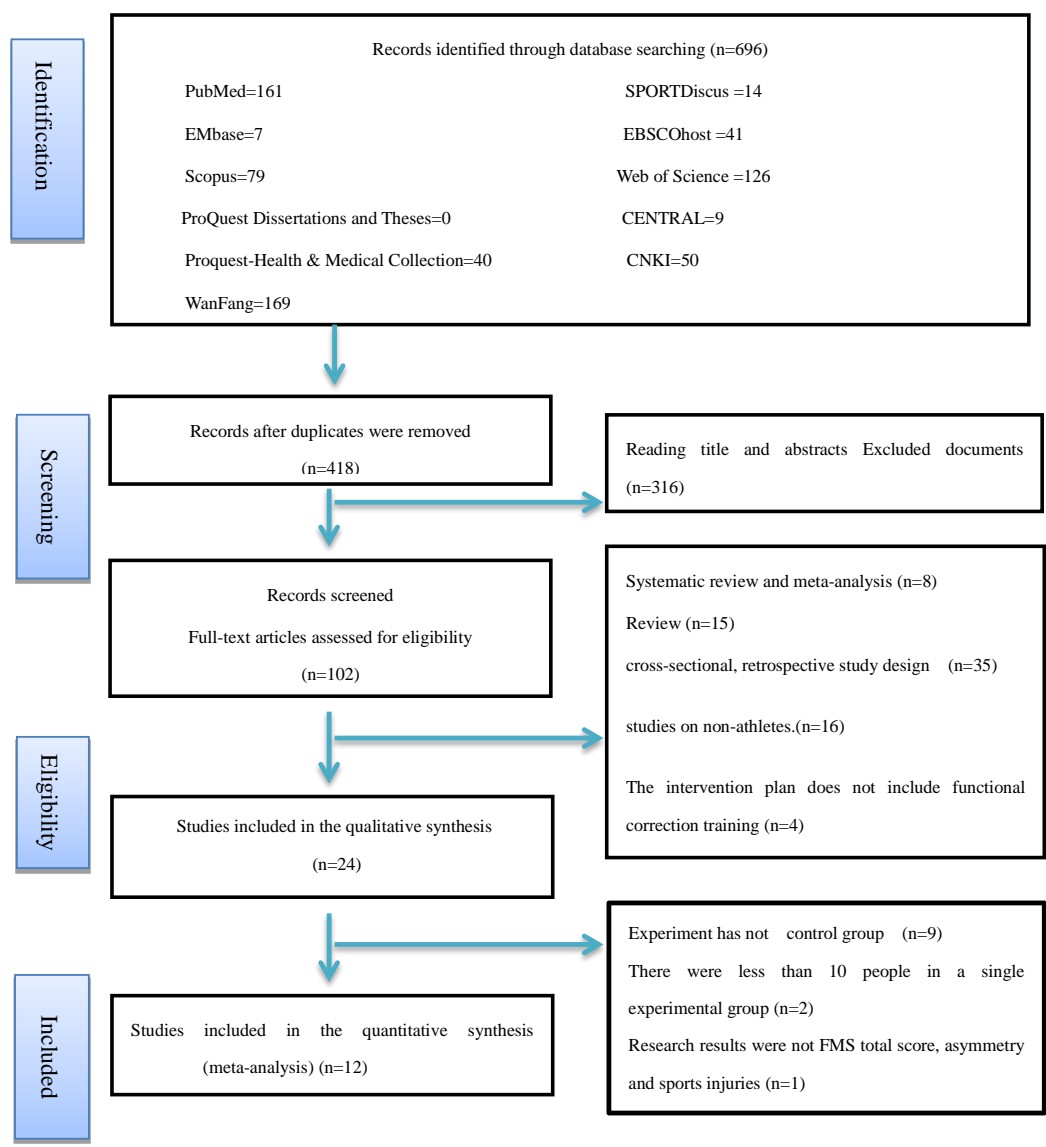

**Figure 1  Flow diagram of the study selection process.**

was twice per week, and the maximum was six times per week. The average intervention frequency was $3.42 \pm 1.39$ times per week. Finally, the shortest duration of each session was 15 min, the longest was 60 min, and the average was $36.36 \pm 16.75$ min. The included studies were published between January 1997 and September 2020. The sports included for research were baseball (*Song et al., 2014*), table tennis (*Kangkang & Zhuhang, 2016*), volleyball (*Xuhua & Ye, 2015*), free kicking (*Bodden, Needham & Chockalingam, 2015*), basketball (*Klusemann et al., 2012*; *Hui & Baoai, 2019*), soccer (*Dinc et al., 2017*; *Campa, Spiga & Toselli, 2019*; *Riela & Bertollo, 2019*; *Schneider et al., 2019*), tennis (*Yildiz, Pinar & Gelen, 2019*), netball (*Kovac, 2018*), and wrestling (*Bayati et al., 2019*). Three studies were not included in the meta-analysis for the following reasons: one was missing data

regarding the total FMS$^{TM}$ scores and sports injury but included data regarding the strength and flexibility of the athletes (*Song et al., 2014*), and two studies had single-group sample sizes fewer than 10 people and very low quality assessment scores (the PEDro quality scale score for physical therapy was only 4 points) (*Kim et al., 2014*; *Armstrong et al., 2019*). Furthermore, some studies used a single-group pre-test design method to perform functional correction training for athletes and found that that they had a positive impact on the FMS$^{TM}$ scores, asymmetric events, and sports injuries (*Kiesel, Plisky & Butler, 2011*; *Lee, Zhang1 & Lee, 2015*; *Garbenyté-Apolinskiené et al., 2018*; *Tejani et al., 2019*; *Boucher et al., 2018*; *Baron et al., 2019*; *Huebner et al., 2019*; *Bayrakdar, Kılın & Boz, 2020*). These studies did not meet the inclusion criteria and were excluded from this report.

The standard function correction program, first classified according to the screening results, and then from basic flexibility to basic stability, and finally retraining the action mode. This procedure includes self-managed trigger point therapy; self-based and partner-based stretching exercises for the major muscle groups; and strength, stability, and flexibility exercises. it uses elastic bands, medicine balls, and foam rollers. In 12 studies, after FMS$^{TM}$ screening, the researchers developed a functional correction training program (including personalized correction training) as an intervention. Some of these studies used mixed interventions, including strength, stability, and jumping (including functional training) of the upper and lower limbs with bare hands or instruments and the Wrestling+ warm-up program (similar functional correction training) (*Klusemann et al., 2012*; *Bayati et al., 2019*). One study did not report whether the intervention plan was supervised by coaches or researchers (*Dinc et al., 2017*). Two studies (*Dinc et al., 2017*; *Bayati et al., 2019*) did not provide any FMS$^{TM}$ assessor qualification information or reliability tests. All experimental groups performed functional correction and conventional training or warm-up activities, and the control groups performed either conventional training or warm-up activities. Two non-randomized trials (*Xuhua & Ye, 2015*; *Hui & Baoai, 2019*) used scoring thresholds to divide the subjects into a high-risk group (total FMS$^{TM}$ score ≤14) and low-risk group (total FMS$^{TM}$ score ≥ 14) prior to the test and interventions (*Kiesel, Plisky & Voight, 2007*). Our report includes four trials.

The numbers of athletes in the experimental and control groups with sports injuries and pattern asymmetry during the intervention period were reported after FMS$^{TM}$ (*Xuhua & Ye, 2015*; *Hui & Baoai, 2019*; *Dinc et al., 2017*; *Bodden, Needham & Chockalingam, 2015*; *Campa, Spiga & Toselli, 2019*; *Kangkang & Zhuhang, 2016*). *Dinc et al. (2017)* did not report the number of athletes with sports injuries; instead, a selection of injuries causing an inability to perform athletic activities for more than three weeks was reported.

### Research bias

The risk of bias was analyzed and a high risk of bias was associated with blinding procedures (Figs. 2 and 3). Participant blinding was only described in one study (*Campa, Spiga & Toselli, 2019*). Four studies included random grouping; however, they did not provide any specific methods. Blinding of the outcome assessors was performed in two studies (*Campa, Spiga & Toselli, 2019*; *Riela & Bertollo, 2019*). The outcome evaluators of the other four studies were not blinded, as repeated measurement reliability, inter-rater reliability, and

**Table 2  Base line characteristics of included studies.**

| References | Sports | n, gender | Age mean (SD) | Experimental group Intervention | Dose | n, gender | Age mean (SD) | Control group Intervention | Dose | Outcomes Measurement item Results between groups |
|---|---|---|---|---|---|---|---|---|---|---|
| *Bodden, Needham & Chockalingam, 2015* United Kingdom | Martial arts | 12, Males | 24.31 ± 4.46 | Corrective exercise program Certified coach implementation | frequency/s,? 4 times/w 8 weeks | 12, Males | 24.13 ± 4.46 | Routine training | 8 weeks | FMS$^{TM}$ scores :EG 15.34 ± 1.43 CG 13.24 ± 0.8 Asymmetry Number of patients Total number EG 2 13 CG 4 12 |
| *Klusemann et al., 2012* Australia; | Basketball | 13, Males and females | 14.6 ± 1 15 ± 1 | Strength, stability and jumping (including functional training) of upper and lower limbs with bare hands or instruments Coach supervision | frequency/s, 60 min 2 times/w 6 weeks | 13, Males and females | 14.6 ± 1 15 ± 1 | Daily training without resistance | 6 weeks | FMS$^{TM}$ scores :EG 16 ± 2 CG 14 ± 1 20-m sprint :EG 3.56 ± 0.21 CG 3.50 ± 0.22 Vertical jump : EG 46 ± 6 CG 44 ± 9 |
| *Campa, Spiga & Toselli, 2019* Italy | Soccer | 32, Males | 15.93 ± 0.4 | Corrective exercise program Professional trainer guidance | 2 times/w 20 weeks | 30, Males | 15. 81 0.63 | Routine training | 20 weeks | FMS$^{TM}$ scores :EG 14.59 ± 0.87 CG 13.13 ± 1.3 Asymmetry Number of patients Total number EG 19 32 CG 22 30 |
| *Yildiz, Pinar & Gelen, 2019* Turkey | Tennis | 10 Males | 9.6 ± 0.7 | Functional training (for problems such as muscle imbalance) Coach supervision | 3 times/w 8 weeks | 10, males | 9.6 ± 0.7 | Routine training | 8 weeks | FMS$^{TM}$ scores : EG 19.3 ± 0.8 CG 10.3 ± 1.6 10-m acceleration: EG 4.44 ± 0.20 CG 3.64 ± 0.3 Counted movement jump: EG 28.9 ± 1.90 CG 22.4 ± 3.6 |
| *Riela & Bertollo, 2019* Italy | Soccer | 15, Males | 23.8 ± 4.6 | Warm up (functional correction training) Professional trainer guidance | 3 times/w 8 weeks | 15, Males | 24.78 ± 4.6 | Regular warm up | 8 weeks | FMS$^{TM}$ scores EG 16.33 ± 0.79 CG 14.21 ± 1.1 |
| *Bayati et al., 2019* Guilan | Wrestling | 12 ? | 16.16 ± 0.7 | Wrestling+" injury prevention program Coach supervision | 3 times/w 12 weeks | 12 ? | 16.41 ± 0.79 | Regular warm up | 12 weeks | FMS$^{TM}$ scores : EG 17.08 ± 0.42 CG 15.47 ± 0.58 |

**Notes.**

(A) EG, experimental group; CG, next step. (B) Amstrong's research results only provide histograms and lack data.

**Table 3   Base line characteristics of included studies.**

| References | Sports | n, gender | Age mean (SD) | Experimental group Intervention | Dose | n, gender | Age mean (SD) | Control group Intervention | Dose | Outcomes Measurement item Results between groups |
|---|---|---|---|---|---|---|---|---|---|---|
| *Dinc et al., 2017* Turkey | Soccer | 24, Males | 16.13 ± 0.38 | Corrective exercise program | 2 times/w 12weeks | 43, Males | 16.42 ± 0.24 | Routine training | 12 weeks | FMS$^{TM}$ scores: EG 16.79 ± 1.61 CG 15.33 ± 1.19 Sports injury (injury stop >3 weeks) Number of patients Total number EG 6 24 CG 31 43 |
| *Song et al., 2014* Korea | Baseball | 31, Males | 17 ± 1.06 | FMS training program | 3 times/w 16 weeks | 31, Males | 16.62 ± 0.94 | Routine training | 16 weeks | Strength (Back Muscle Strength) : EG 144.93 ± 20.67 CG 137.74 ± 20.5 Strength squat (1RM): EG 161.08 ± 35.06 CG 129.68 ± 26.82 |
| *Schneider et al., 2019* Germany | Soccer | 23, Males | 11.87 ± 0.87 | Individualized multimodal training intervention on warm up Coach supervision | 2 times/w 12 weeks | 22, Males | 10.84 ± 1.18 | Regular soccer practice | 12 weeks | FMS$^{TM}$ scores : EG 14.30 ± 143 CG 13.16 ± 2.44 |
| *Kangkang & Zhuhang, 2016* China | Table tennis | 20, Males and females | ? | Pre-class function plan × 4 + personalized correction training × 1 (supervised by author and fitness coach) | 5 times/w 6 weeks | 20, Males and females | ? | Routine training | 6 weeks | FMS$^{TM}$ scores: EG 15.15 ± 1.27 CG 13.15 ± 1.35 Asymmetry Number of patients Total number EG 1 20 CG 9 20 |
| *Hui & Baoai, 2019* China | Basketball | High-risk 8, males Low risk 8, males | 21.75 ± 1.28 21.50 ± 0.76 | Dynamic stretching and personalized correction training | 6 times/w 8 weeks | High-risk 8, males Low risk 8, males | 21.78 ± 1.48 21.71 ± 1.49 | Routine training | 8 weeks | FMS$^{TM}$ scores (High-risk group): EG 14.00 ± 1.31 CG 12.44 ± 1.01 (Low risk group): EG 16.25 ± 1.75 CG 15.42 ± 0.78 High-risk Number of patients Total number EG 3 8 CG 5 8 Low-risk EG 1 8 CG 2 8 |
| *Kovac, 2018* South Africa | Netball | 10, Females | 20 ± 1.5 | Corrective exercise program instructed and supervised by the researcher. | 3 times/ 6 weeks | 19, Females | 19.8 ± 1.5 | Routine training | 6 weeks | FMS$^{TM}$ scores : EG 14.55 ± 1.6 CG 13.55 ± 2.4 Drop vertical jump: EG 2.155 ± 0.95 CG 1.9 ± 0.86 |

**Notes.**

(A) EG, experimental group; CG, next step. (B) Amstrong's research results only provide histograms and lack data.

Peer J

**Table 4  Base line characteristics of included studies.**

| References | Sports | n, gender | Age mean (SD) | Experimental group Intervention | Dose | n, gender | Age mean (SD) | Control group Intervention | Dose | Outcomes Measurement item Results between groups |
|---|---|---|---|---|---|---|---|---|---|---|
| *Armstrong et al., 2019* USA | Basketball | 6, Males | 20.04 ± 1.4 | Corrective exercise program | 4 times/w 4 weeks | 7, Males | 20.04 ± 1.4 | Pre-practice dynamic warm-up | 4 times/w 4 weeks | Incomplete data |
| *Xuhua & Ye, 2015* China | Volleyball | High-risk 15, females Low risk 13, females | 20.92 ± 3.26 21.47 ± 3.16 | Rehabilitation physical training (correction training) | 6 times/ 6 weeks | High-risk 14, females Low risk 12, females | 21.56 ± 3.58 21.20 ± 3.32 | Routine training | 6 weeks | FMS$^{TM}$ scores (High-risk group) EG 14.80 ± 1.21 CG 12.21 ± 1.05 (Low risk group): EG 17.23 ± 2.05 CG 15.33 ± 1.30 High-risk Number of patients Total number EG 3 15 CG 8 14 Low-risk EG 1 13 CG 2 12 Squat (High-risk) :EG 115 ± 12.11 CG 112 ± 18.78 (low-risk) :EG 118 ± 6.15 CG 115 ± 18.7 |
| *Kim et al., 2014* Korea | Javelin | 4, Males 2, Females | Males 22 ± 1.15 Females 22 ± 1.41 | Weight, Javelin specific, core, FMS training Performed by researchers | ?/8weeks | 2, Males 2, females | Males 26 ± 4.24 Females 26.5 ± 1.41 | Routine training | 8 weeks | Difference CG-EG FMS score (points): CG 0.30 ± 1.07 EG-1.03 ± 1.37 throwing performances: CG 9.6 ± 1.10 EG5.8 ± 2.64 |

**Notes.**

(A) EG, experimental group; CG, next step. (B) Amstrong's research results only provide histograms and lack data.
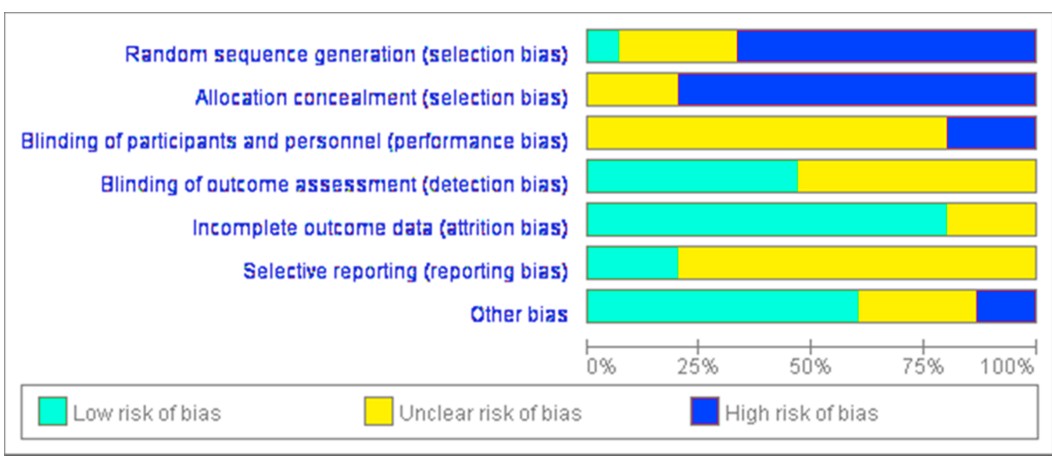

**Figure 2** Assessment of bias risk for included studies (risk of bias graph).

high-precision professional electronic instruments were used to record data (*Song et al., 2014*; *Kim et al., 2014*; *Klusemann et al., 2012*; *Yildiz, Pinar & Gelen, 2019*). The selection bias ratings remained unclear due to either insufficient or unclear information. A low risk of bias attributable to the blinding of outcome assessment, reporting, and other bias was observed throughout the studies.

We used the PEDro physical therapy quality scale to evaluate article quality and the primary difference between randomized and non-randomized trials (Table 5). Some studies included athletes who were randomly divided into groups in a blinded manner (*Bodden, Needham & Chockalingam, 2015*; *Kovac, 2018*; *Campa, Spiga & Toselli, 2019*; *Riela & Bertollo, 2019*). Some studies did not conduct random grouping of athletes, therefore, there was no score for this item. In some studies, the coaches, raters, and participants were not blinded, therefore, scores were not obtained for questions related to those items. Among the 15 studies, the average score was 5.5 with an overall quality of the literature of average.

## Result integration

We verified the effects of functional correction training on sports injuries of athletes based on the sports injury RR, total FMS$^{TM}$ score, and functional pattern asymmetry. There was no heterogeneity in the hazard ratio of the influence of functional correction training on athletes' sports injuries (RR, 0.39; 95% CI [0.24–0.65]; $Z = -3.57$; $P = 0.0003$; $I^2 = 0.0\%$) (Fig. 4); therefore, the fixed-effects model was used to combine the effect sizes. The incidence of sports injuries in the experimental group was lower than that of the control group, and the injury risk in the experimental group decreased by 60%.

The effect sizes were combined to measure the influence of functional correction training on the total FMS$^{TM}$ scores (MD, 1.72; 95% CI [1.50–1.93]; $Z = 15.53$; $P < 0.0001$; $I^2 = 2.6\%$) (Fig. 5). As there was low heterogeneity, the fixed-effects model was used to combine the effect sizes. All results had large effect sizes with significant differences as compared

**Table 5  Quality assessment results of included studies using the PEDro quality scales.** The purpose of the PEDro scale is to help the users of the PEDro database rapidly identify which of the known or suspected randomized clinical trials (i.e., RCTs or CCTs) archived in the database are likely to be internally valid (criteria 2–9), and could have sufficient statistical information to make their results interpretable (criteria 10–11). An additional criterion (criterion 1) that relates to the external validity (or "generalizability" or "applicability" of the trial) has been retained so that the Delphi list is complete, but this criterion will not be used to calculate the PEDro score reported on the PEDro web site.

| Author | 1 | 2 | 3 | 4 | 5 | 6 | 7 | 8 | 9 | 10 | 11 | scale | PEDro Scoring item |
|---|---|---|---|---|---|---|---|---|---|---|---|---|---|
| *Dinc et al. (2017)* | Y | N | N | Y | N | N | N | Y | Y | Y | Y | 5 | 1. eligibility criteria were specified |
| *Bodden, Needham & Chockalingam (2015)* | Y | N | N | Y | N | N | ? | Y | Y | Y | Y | 5 | 2. subjects were randomly allocated to groups |
| *Campa, Spiga & Toselli (2019)* | Y | Y | N | Y | N | N | Y | Y | Y | Y | Y | 7 | 3. allocation was concealed |
| *Kovac (2018)* | Y | Y | N | Y | N | N | N | Y | Y | Y | Y | 6 | 4. the groups were similar at baseline regarding the most important prognostic indicators |
| *Riela & Bertollo (2019)* | Y | Y | ? | Y | N | N | Y | Y | Y | Y | Y | 7 | 5. there was blinding of all subjects |
| *Song et al. (2014)* | Y | N | N | Y | N | N | N | Y | Y | Y | Y | 5 | 6. there was blinding of all therapists who administered the therapy |
| *Schneider et al. (2019)* | Y | N | N | Y | N | N | ? | Y | Y | Y | Y | 5 | 7. there was blinding of all assessors who measured at least one key outcome |
| *Xuhua & Ye (2015)* | Y | N | N | Y | N | N | ? | Y | Y | Y | Y | 5 | 8. measures of at least one key outcome were obtained from more than 85% of the subjects initially allocated to groups |
| *Kangkang & Zhuhang (2016)* | Y | N | N | Y | N | N | N | Y | Y | Y | Y | 5 | 9. all subjects for whom outcome measures were available received the treatment or control condition as allocated or, where this was not the case, data for at least one key outcome was a analyses by "intention to treat |
| *Klusemann et al. (2012)* | Y | Y | N | Y | N | N | ? | N | Y | Y | Y | 5 | 10. the results of between-group statistical comparisons are reported for at least one key outcome |
| *Bayati et al. (2019)* | Y | N | N | Y | N | N | ? | Y | Y | Y | Y | 5 | 11. the study provides both point measures and measures of variability for at least one key outcome |
| *Yildiz, Pinar & Gelen (2019)* | Y | N | ? | Y | N | N | ? | Y | Y | Y | Y | 5 | |
| *Hui & Baoai (2019)* | Y | N | N | Y | N | N | N | Y | Y | Y | Y | 5 | |
| *Kim et al. (2014)* | Y | N | N | ? | N | N | N | Y | Y | Y | Y | 4 | |
| *Armstrong et al. (2019)* | Y | Y | N | ? | N | N | N | Y | Y | N | Y | 4 | |

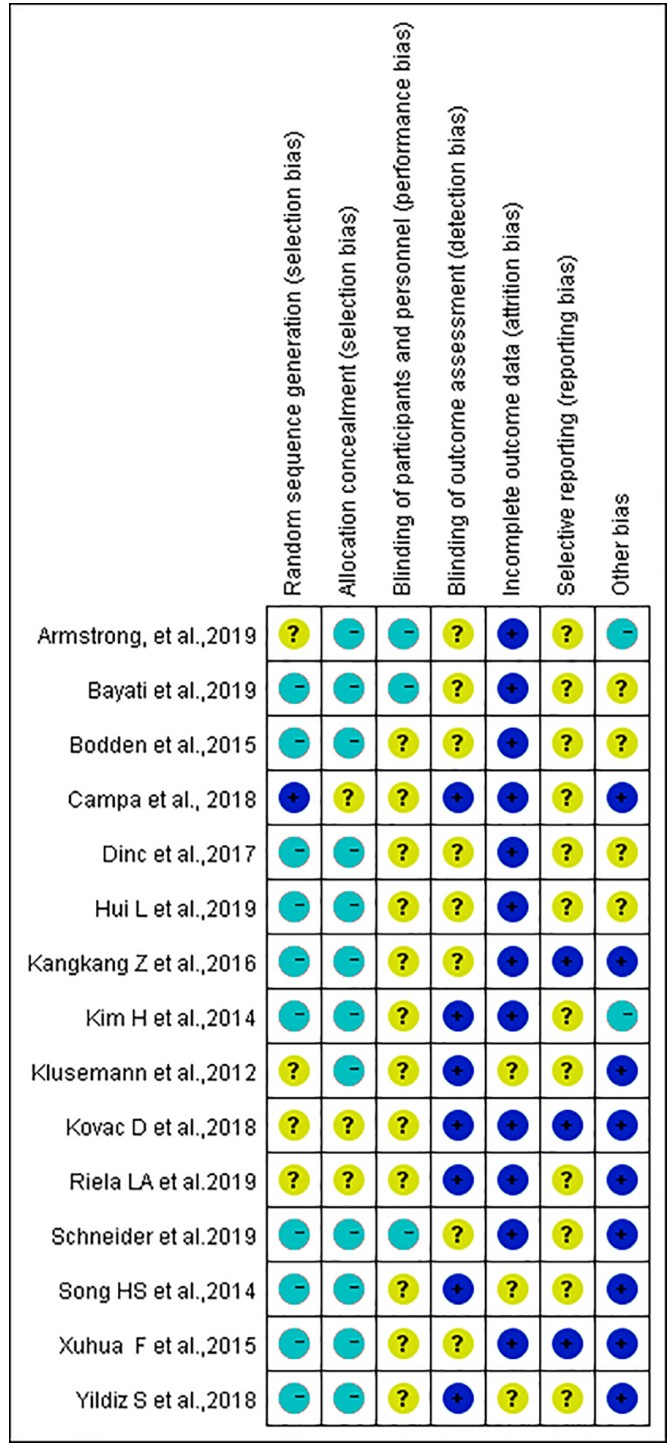

**Figure 3   Assessment of bias risk for included studies (risk of bias summary).**

with those of the control group and the functional patterns of athletes were optimized according to Cohen's interpretation standard.

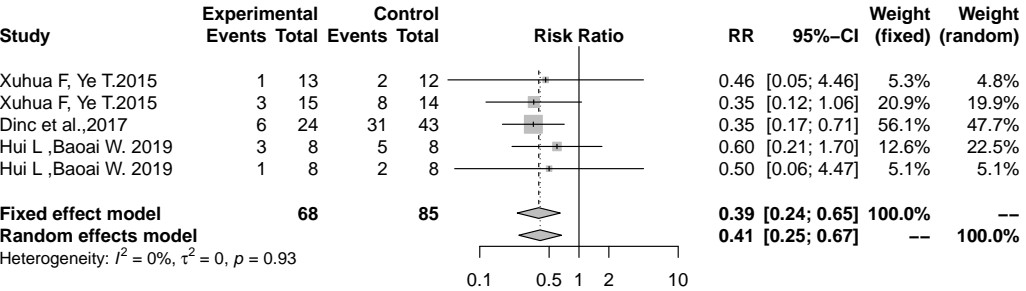

**Figure 4** Forest plot of athletes' sports injuries.

The hazard ratio of the influence of functional correction training on the pattern of asymmetry of athletes showed large heterogeneity (RR, 0.45; 95% CI [0.13–1.50]; $Z = -1.3; P = 0.19; I^2 = 65.2\%$) (Fig. 6). Therefore, the random effects model was used to combine the effects and no significant difference was observed when compared with the control group. One study considered that the age of the athlete may explain heterogeneity in the prospective prediction of injury risk by FMS[TM] (*Moore et al., 2019*) and reported that asymmetry determined by FMS[TM] is more useful for evaluating the injury risk of senior athletes. Another study included soccer players aged 15.89 ± 0.53 years (*Campa, Spiga & Toselli, 2019*), and two studies included adult-free combat athletes (*Bodden, Needham & Chockalingam, 2015*) and a national table tennis team (*Kangkang & Zhuhang, 2016*). The incidence rates of model asymmetry for adults were lower than that of the younger soccer players. Therefore, our research results are in line with their results. However, a subgroup analysis to determine the source of heterogeneity was impossible as only three cases were included in the sample. Additionally, the total FMS[TM] score was not necessarily improved, and a score of 21 was not the goal. Instead, the focus was the identification of asymmetries (*Cook et al., 2014b*). Further studies are needed to explore the influence of functional correction training on the model asymmetry of athletes.

## Publication bias and sensitivity analysis

The publication bias associated with the influence of functional correction training on athletes' total FMS[TM] scores was not significant (Egger's linear regression, $t = -0.096$; $df = 11; P = 0.92 > 0.05$) (Fig. S1). The sensitivity analysis indicated that the hazard ratios for athletic injuries and total FMS[TM] scores after functional correction training were consistent with those without stratification, with very robust results (Figs. S2–S5). The results of the sensitivity analysis of athletes' asymmetry were slightly different than those before stratification and were not sufficiently stable.

## Level of evidence

The included studies were non-randomized controlled trials, and the level of evidence using GRADE instruments was low Tables 6 and 7). Altogether, these studies provided a very low level of evidence of the injury risk ratio and asymmetry model of the athlete. The other prevalent outcome showed a moderate level of evidence of athletes' total FMS[TM] scores.

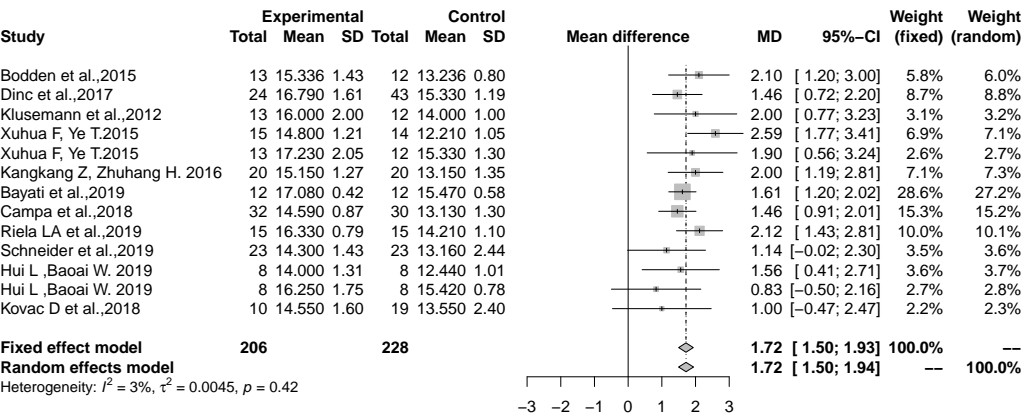

**Figure 5** Forest plot of the effect size of the athletes' total FMS^TM score.

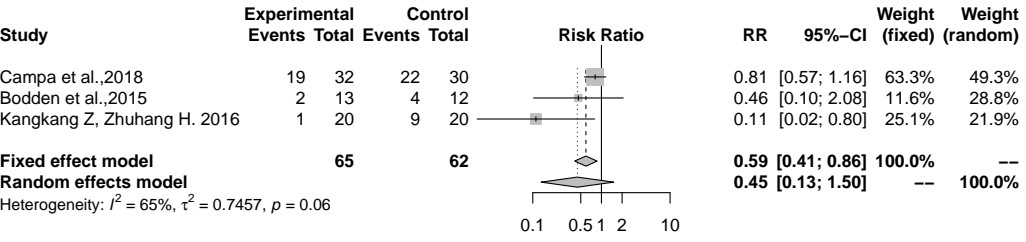

**Figure 6** Forest plot of the athletes' asymmetry functional patterns.

## DISCUSSION

Our review explored the influence of functional correction training based on FMS^TM on the sports injury risk of athletes. The results showed that the injury risk of the experimental group was reduced by 60% after functional correction training, the effect on the total scores of FMS^TM was large, and significantly different from that of the control group. Results of the sensitivity analysis were very robust, and the possibility of publication bias influencing the athletes' total FMS^TM scores was very low.

A previous review had no consistent conclusion regarding the total FMS^TM score and the risk of subsequent injuries for athletes and mixed populations (*Dorrel et al., 2015*; *Moran et al., 2017*; *Bonazza et al., 2017*; *Santos Bunn, Rodrigues & Da Silva, 2019*; *Moore et al., 2019*). This review summarized the functional correction training after FMS^TM with the RR for athletes' injuries, total FMS^TM scores, and asymmetry. We provided standardized evidence and clarified that functional correction training after FMS^TM can effectively enhance the functional patterns of athletes.

The positive effects of functional correction training after FMS^TM on sports injuries may be due to several factors. First, using FMS^TM, athletes may discover weaknesses and perform corrections by focusing on their trunk pillar strength, joint flexibility, and joint stability to ensure effectiveness. Second, intervention training includes myofascial therapy,

Chen et al. (2021), *PeerJ*, DOI 10.7717/peerj.11089

**Table 6 Summary of findings.**

Effect of Functional Correction Training on Injury Risk of Athletes: A Systematic Review and Meta-analysis
Patient or population: athletes                                             Setting : sports injury
Intervention: functional correction training                               Comparison:conventional training

| Outcomes | Anticipated absolute effects * (95% CI) | | Relative effect (95% CI) | No. of participants (studies) | Certainty of the evidence (GRADE) | Comments |
|---|---|---|---|---|---|---|
| | Risk with conventional training | Risk with functional correction training | | | | |
| **Study population** | | | | | | |
| | **Study population** | | | | | |
| sports injury risk of athletes (injury risk) follow up: mean 6–12 weeks | 565 per 1,000 | **222 per 1,000** (135 to 366) | **RR 0.3932** (0.2386 to 0.6482) | 153 (5 observational studies) | ⊕○○○ VERY LOW [a,b,c,d] | The injury risk ratio of athletes after functional correction training was 0.3932 RR (95% CI, 0.2386–0.6482; Z = −3.57; P = 0.0003; I²=0.0%). It was found that functional correction training could reduce the injury risk by 60% in the experimental groups as compared with the control groups. |
| | 0 per 1,000 | **0 per 1,000** (0 to 0) | | | | |
| **Study population** | | | | | | |
| New outcome (model asymmetry of athletes) assessed with: Functional movement screen follow up: mean 6-20 weeks | 565 per 1,000 | **252 per 1,000** (75 to 849) | **RR 0.4460** (0.1323 to 1.5033) | 127 (3 observational studies) | ⊕○○○ VERY LOW [c,d,e,f] | The hazard ratio of the influence of functional correction training on the pattern of asymmetry of athletes showed large heterogeneity (RR, 0.446; 95% CI, 0.1323–1.5033; Z = −1.3; P = 0.1928; I² = 65.2%). and no significant difference was observed when compared with the control group. |
| | **Moderate** | | | | | |
| | 0 per 1,000 | **0 per 1,000** (0 to 0) | | | | |
| Total FMS score of athlete assessed with: FunctionalMovement Screen Scale from: 0 to 21 follow up: range 6 weeks to 20 weeks | The mean total FMS score of athlete was **13.89** MD | MD **1.7165 MD higher** (1.4999 higher to 1.9333 higher) | – | 434 (13 observational studies) | ⊕⊕⊕○ MODERATE [g,h] | The influence of functional correction training on the athletes' total FMS^TM scores is 1.7165 MD (95% CI, 1.4999–1.9330; Z=15.53; P<0.0001; I² =2.6%), indicating effective improvement of athletes' functional patterns. |

*The risk in the intervention group (and its 95% confidence interval) is based on the assumed risk in the comparison group and the **relative effect** of the intervention (and its 95% CI).
**CI:** Confidence interval; **RR:** Risk ratio; **MD:** Mean difference

**GRADE Working Group grades of evidence**
**High certainty:** We are very confident that the true effect lies close to that of the estimate of the effect
**Moderate certainty:** We are moderately confident in the effect estimate: The true effect is likely to be close to the estimate of the effect, but there is a possibility that it is substantially different
**Low certainty:** Our confidence in the effect estimate is limited: The true effect may be substantially different from the estimate of the effect
**Very low certainty:** We have very little confidence in the effect estimate: The true effect is likely to be substantially different from the estimate of effect

**Notes.**

Explanations

[a] All included studies were not randomized.

[b] Researchers may have different definition of sports injury/injuries.

[c] Sample size was small. According to the graph in the GRADE guidelines: 6. Rating the quality of evidence—imprecision, set RRR=30%, both injury risk ratio and asymmetry movement patterns of athletes that event rate of the control group was 0.56, at least 500–1,000 samples were required.

[d] The publication bias test was not completed because the sample sizes used to determine the sports injury risk and model asymmetry were fewer than 10.

[e] Only one study implemented randomization.

[f] The hazard ratio of the influence of functional correction training on patterns of athletes' asymmetry had large heterogeneity (RR, 0.446; 95% CI [0.1323–1.5033]; $z = -1.3$; $P = 0.1928$; $I2 = 65.2\%$).

[g] 66% of subjects were not randomly allocated to a group.

[h] The influence of functional correction training on the athletes' total FMS^TM scores was 1.7165 (95% CI [1.4999–1.9330]; $Z = 15.53$; $P < 0.0001$; $I^2 = 2.6\%$), Confidence interval exceeded 1.

**Table 7 GRADE evidence profile.**

| No of studies | Study design | Certainty assessment | | | | | No. of patients | | Relative (95% CI) | Effect | Certainty | Importance |
|---|---|---|---|---|---|---|---|---|---|---|---|---|
| | | Risk of bias | Inconsistency | Indirectness | Imprecision | Other considerations | functional correction training | conventional training | | Absolute (95% CI) | | |
| colspan | | | | | | **sports injury risk of athletes (follow up: mean 6-12 weeks)** | | | | | | |
| 5 | observational studies | serious [a],[b] | not serious | not serious | serious [c] | publication bias strongly suspected strong association all plausible residual confounding would reduce the demonstrated effect [d] | 14/68 (20.6%) | 48/85 (56.5%) 0.0% | RR 0.3932 (0.2386 to 0.6482) | 343 fewer per 1,000 (from 430 fewer to 199 fewer) 0 fewer per 1,000 (from 0 fewer to 0 fewer) | ⊕ooo VERY LOW | CRITICAL |
| colspan | | | | | | **New outcome (follow up: mean 6-20 weeks; assessed with: Functional movement screen)** | | | | | | |
| 3 | observational studies | very serious [e] | serious [f] | not serious | serious [c] | publication bias strongly suspected all plausible residual confounding would reduce the demonstrated effect [d] | 22/65 (33.8%) | 35/62 (56.5%) 0.0% | RR 0.4460 (0.1323 to 1.5033) | 313 fewer per 1,000 (from 490 fewer to 284 more) 0 fewer per 1,000 (from 0 fewer to 0 fewer) | ⊕ooo VERY LOW | CRITICAL |
| colspan | | | | | | **Total FMS score of athlete (follow up: range 6 weeks to 20 weeks; assessed with: Functional Movement Screen; Scale from: 0 to 21)** | | | | | | |
| 13 | observational studies | serious [g] | not serious | not serious | serious [h] | very strong association all plausible residual confounding would reduce the demonstrated effect | 206 | 228 | – | MD **1.7165 MD higher** (1.4999 higher to 1.9333 higher) | ⊕⊕⊕o MODERATE | IMPORTANT |

**Notes.**

CI, Confidence interval; RR, Risk ratio; MD, Mean difference.

Explanations

[a] All included studies were not randomized.

[b] Researchers may have different definition of sports injury/injuries.

[c] Sample size was small. According to the graph in the GRADE guidelines: 6. Rating the quality of evidence—imprecision, set RRR=30%, both injury risk ratio and asymmetry movement patterns of athletes that event rate of the control group was 0.56, at least 500-1,000 samples were required.

[d] The publication bias test was not completed because the sample sizes used to determine the sports injury risk and model asymmetry were fewer than 10.

[e] Only one study implemented randomization.

[f] The hazard ratio of the influence of functional correction training on patterns of athletes' asymmetry had large heterogeneity (RR, 0.446; 95% CI [0.1323–1.5033]; $z = -1.3$; $P = 0.1928$; $I2 = 65.2\%$).

[g] 66% of subjects were not randomly allocated to a group.

[h] The influence of functional correction training on the athletes' total FMS$^{TM}$ scores was 1.7165 (95% CI [1.4999–1.9330]; $Z = 15.53$; $P < 0.0001$; $I^2 = 2.6\%$), Confidence interval exceeded 1.

dynamic stretching, core stability training, resistance strength training, and combined neuromuscular training. This may improve the imbalance of the muscle groups and the energy transmission effect of the body's kinetic chain (*Cook, 2011*; *Cook et al., 2014a*; *Cook et al., 2014b*). Third, functional correction training includes core stability exercises. Enhancing core stability through exercise is common to musculoskeletal injury prevention programs. Core stabilization relies on instantaneous integration among passive, active, and neural control subsystems. Neuromuscular control is critical in coordinating this complex system for dynamic stabilization (*HuxelBliven & Anderson, 2013*). Fourth, these interventions aim to stimulate the activation of the muscles of the natural nucleus to improve the relationship between the main muscular function and the fundamental movement (*Cook & Fields, 1997*; *Kiesel, Plisky & Butler, 2011*). Additionally, an 8-week program with the foam roll has been reported as effective in increasing range of motion in the stand and reach test (*Junker & Stöggl, 2019*). The theory that functional correction training programs should consist of functional movements related with core stability and shoulder and hamstring flexibility improvement is supported by a study that reported improved strength and flexibility in 62 elite male high school baseball players after participating in a correction training program (*Song et al., 2014*). Therefore, functional correction training may effectively reduce the risk of sports injury.

This was the first study to evaluate the impact of functional correction training after FMS$^{TM}$ on athletes' sports injury risk by including non-randomized controlled trials. Grade B evidence indicates that functional correction training based on FMS$^{TM}$ could improve athletes' functional patterns and Grade D evidence indicates that functional correction training may reduce the risk of sports injuries in athletes. The evidence found in this review is reliable and significant for evidence-based clinical practice.

## Strengths and limitations

Our review had some limitations. First, some relevant literature may have been overlooked despite a search of ten online databases. Second, the methodological limitations of this review include the small sample sizes evaluated within the retained studies, no differentiation among sports, allocation concealment, and evaluator blindness, which may have resulted in an overestimation of the effects of the intervention. Third, because the sample sizes used to determine the sports injury risk and model asymmetry were fewer than 10, the publication bias test was not completed. Fourth, some studies did not define whether the coaches supervised or corrected the training quality or whether joint intervention was used. Fifth, FMS$^{TM}$ was limited by its inability to test a single construct from a composite set of scores. The total FMS$^{TM}$ score of our study was only used to show whether the functional model could be improved. Thus, our findings should be carefully interpreted.

## CONCLUSIONS

Grade B evidence indicates that functional correction training based on FMS$^{TM}$ could improve the functional patterns of athletes, and Grade D evidence indicates that it may reduce the risk of sports injury. The true effect is likely to be different from the estimate

of effect. Therefore, further studies are needed to explore the influence of functional correction training on the injury risks of athletes.

## ACKNOWLEDGEMENTS

We would like to thank Professors Li Shuping, Huang Zhijian, and Yu Lianghua for their guidance and suggestions in the process of writing the paper. We would also like to thank Editage for their editing work for this article.

### Funding

This study was financed by grants from the Teaching Research Project of Hubei University (No.201978), and the Open Project of the Key Laboratory of the State Sports General Administration (No. 2017C17). The funders had no role in study design, data collection and analysis, decision to publish, or preparation of the manuscript.

### Grant Disclosures

The following grant information was disclosed by the authors:
Teaching Research Project of Hubei University: 201978.
Open Project of the Key Laboratory of the State Sports General Administration: 2017C17.

### Competing Interests

The authors declare there are no competing interests.

### Author Contributions

- Junxia Chen conceived and designed the experiments, performed the experiments, analyzed the data, prepared figures and/or tables, authored or reviewed drafts of the paper, and approved the final draft.
- Chunhe Zhang conceived and designed the experiments, performed the experiments, analyzed the data, authored or reviewed drafts of the paper, and approved the final draft.
- Sheng Chen and Yuhua Zhao performed the experiments, analyzed the data, prepared figures and/or tables, and approved the final draft.
- performed the experiments, analyzed the data, prepared figures and/or tables, and approved the final draft.

### Data Availability

The raw data collected in this systematic review and meta-analysis are available in the Supplementary Files.

### Supplemental Information

Supplemental information for this article can be found online at http://dx.doi.org/10.7717/peerj.11089#supplemental-information.

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
