# Peer review of "Effects of functional correction training on injury risk of athletes: a systematic review and meta-analysis"

_PeerJ, doi:10.7717/peerj.11089_

## Round 0.1 · original submission · Major Revisions

I have read your manuscript and the reviewers' comments, and I think your work has scientific merit to be published in PeerJ. However, there are some issues which you must address in a revised version of the text.

·

Basic reporting

 In some phrases some supporting references are missing. This problem is repeatedly seen in whole of the manuscript. Examples: lines 89 and 94
 Please explain the importance of research. Lines 130-138
 Explain more about FMS before the second paragraph and previous studies. Line 95
 In the discussion section, please use more theoretical foundations to better interpret the results.

Experimental design

Well done

Validity of the findings

Well done

Additional comments

Title: Effects of functional correction training on injury risk of athletes - a systematic review and meta-analysis.

This study investigated the Effects of functional correction training on injury risk of athletes in a systematic review and meta-analysis study. Results indicated that the functional correction training can reduce the injury risk. Furthermore the functional correction training can improve the FMS™ scores. Even though the subject is very relevant and current, I have some concerns that I would like to be addressed before considering this paper for publication.

Reviewer 2 ·

Basic reporting

English proofread is absolutely necessary before the paper can be recommended for publication. This include the figures as well.

Otherwise, the reporting is more or less clear and concise. References are sufficient. A hypothesis should be included in the Introduction section.

Figures are well prepared (apart from grammatical mistakes).

Experimental design

Research question is well defined and relevant. The results of the literature review are important for practitioners and researchers. The search strategy was well designed.

The only major comment I have regarding the experimental design is related to inclusion/exclusion criteria: It is unclear what was the requirement for the control group. From what is written, I assume that the control group athletes continued with their regular training? Does this mean that your comparisons analyze regular training + corrective training versus regular training only? Or regular training vs. corrective training? I would also strongly suggest that the authors structure these criteria according to the PICOS tool for better clarity.

Validity of the findings

No comments.

Additional comments

Some specific suggestions for improvements:

ABSTRACT
I would omit the ‘’non-randomized’’ in the abstract background section. This could imply that you purposefully include only ‘’non-randomized’’ trials. You nicely state later that both randomized and non-randomized trials were considered. Maybe, instead of the background, you could start the results section with ‘’12 non-randomized trials were included into the meta-analysis’’.
Line 58. It is not clear what 1.7165 stand for. Is it the mean difference?
Line 62. Insert a full stop after ‘’athletes’’
Line 62-63. Either insert a full stop after ‘’effect’’, or write ‘’therefore’’ in small caps.

INTRODUCTION
Line 95. What do you mean by function-based exercise? Please explain more clearly in the manuscript.
Lines 130-138. What was your hypothesis after analyzing the available literature?

METHODS
Line 186. Bilateral instead of left and right.
Line 190-192. Please revise this sentence. It is unclear what you mean.
Line 225. You need to elaborate more on the sensitivity analysis. What exactly was done?

RESULTS

Line 252. Session, not lesson.
Line 328. Random, not randomized
Lines 342-347. This section should be integrated into the Discussion, not Results.

DISCUSSION
Line 377-378. Your review does not show WHY the training was effective. Therefore, you can only speculate about the reasons, and thereby, sentences such as this one are not appropriate.

·

Basic reporting

Tables and figures are clear for me, and looks attractive for readers. The references should be corrected. Because some of the citations cannot be found in reference list.
Authors should also consider about proper structure of this article, because some of the pages are double.

Experimental design

In my opinion some of sentences (highlighted in the text) should be explained more, for better understanding of international community. Methodoly are clearly descibed with the help of figures and tables.

Validity of the findings

No comment.

Additional comments

I believe that the article provides a good overview and summary of the FMS.
Readers are able to see problems occurring making researches, and to plan their own with better rigour.
This paper should be good base for preparing potential research with the use of FMS.
I hope if authors make cosmetic changes, this paper would be valuable for scientific world.

---

## Round 0.2 · Minor Revisions

Still pending some minor corrections which you should address in a new revised version of the text.

·

Basic reporting

acceptance

Experimental design

OK

Validity of the findings

OK

Additional comments

OK

Reviewer 2 ·

Basic reporting

No comments left.

Experimental design

No comments left.

Validity of the findings

No comments left.

Additional comments

Regarding the contents of the paper, I am satisfied with the modifications that the authors have made. Well done, congratulations. I still think the paper could be somewhat improved in terms of grammar and style. For instance, there are way too many decimal places at several points in results.

---

## Round 0.3 · accepted · Accept

All the reviewers' concerns have been correctly addressed.

Reviewer 2 ·

Basic reporting

OK

Experimental design

OK

Validity of the findings

OK

Additional comments

All comments have been addressed, well done.